# MAPS: pathologist-level cell type annotation from tissue images through machine learning

Muhammad Shaban[1,2,3,4,10], Yunhao Bai[5,10], Huaying Qiu[6,10], Shulin Mao[6], Jason Yeung[6], Yao Yu Yeo [6], Vignesh Shanmugam[1,4], Han Chen[5], Bokai Zhu[5], Jason L. Weirather[3,7], Garry P. Nolan [5], Margaret A. Shipp [8], Scott J. Rodig[1,8], Sizun Jiang [4,6,9,11] & Faisal Mahmood [1,2,3,4,11]

Highly multiplexed protein imaging is emerging as a potent technique for analyzing protein distribution within cells and tissues in their native context. However, existing cell annotation methods utilizing high-plex spatial proteomics data are resource intensive and necessitate iterative expert input, thereby constraining their scalability and practicality for extensive datasets. We introduce MAPS (Machine learning for Analysis of Proteomics in Spatial biology), a machine learning approach facilitating rapid and precise cell type identification with human-level accuracy from spatial proteomics data. Validated on multiple in-house and publicly available MIBI and CODEX datasets, MAPS outperforms current annotation techniques in terms of speed and accuracy, achieving pathologist-level precision even for typically challenging cell types, including tumor cells of immune origin. By democratizing rapidly deployable and scalable machine learning annotation, MAPS holds significant potential to expedite advances in tissue biology and disease comprehension.

The precise delineation of cellular subtypes is crucial for elucidating structural and functional intricacies of biological tissues, within their native context. Compared to conventional low-plex imaging methods, recent advances in high-plex spatial proteomics techniques, such as MIBI, CODEX, cycIF, and IMC, allows for interrogation of 40–60 proteomic markers within a single tissue section[1-7]. These approaches offer invaluable insights into protein expression and distribution within cellular and tissue architectures for phenotypic and functional investigations, and are broadly applicable to fields such as cancer-immune and host-pathogen interactions[8-10]. Iterative cyclical methods, such as CODEX and cycIF, achieve multiplexity through multiple cycles of staining, imaging, and bleaching/stripping of the labeling molecules using fluorescent microscopy methods and off-the-shelf reagents. However, these approaches may face barriers related to tissue degradation, difficulties in image registration, and epitopes loss during the cycling process. Mass spectroscopy-based methods, such as MIBI and IMC, are a different imaging modality requiring specialized instruments and custom conjugation of antibodies with isotopes. In this case, all the markers may be acquired simultaneously to directly reconstruct multiplexed images for downstream analysis.

[1]Department of Pathology, Brigham and Women's Hospital, Harvard Medical School, Boston, MA, USA. [2]Department of Pathology, Massachusetts General Hospital, Harvard Medical School, Boston, MA, USA. [3]Cancer Data Science Program, Dana-Farber Cancer Institute, Boston, MA, USA. [4]Broad Institute of Harvard and MIT, Cambridge, MA, USA. [5]Department of Pathology, Stanford University School of Medicine, Stanford, CA, USA. [6]Center for Virology and Vaccine Research, Beth Israel Deaconess Medical Center, Harvard Medical School, Boston, MA, USA. [7]Center for Immuno-oncology, Dana-Farber Cancer Institute, Harvard Medical School, Boston, MA, USA. [8]Department of Medical Oncology, Dana-Farber Cancer Institute, Harvard Medical School, Boston, MA, USA. [9]Department of Pathology, Dana Farber Cancer Institute, Boston, MA, USA. [10]These authors contributed equally: Muhammad Shaban, Yunhao Bai, Huaying Qiu. [11]These authors jointly supervised this work: Sizun Jiang, Faisal Mahmood. ✉e-mail: sjiang3@bidmc.harvard.edu; faisalmahmood@bwh.harvard.edu

While these highly multiplexed images may provide new insights into biological processes, they also pose challenges in data processing, including the need for automated pipelines to distill the information from each single cell. For example, accurate cell type annotation presents formidable challenges, stemming primarily from constraints in highly precise cell segmentation[11], lateral spillover of markers in tightly packed tissues[12], presence of tissue-level and patient-level variability, and heterogeneous expression patterns[9,10,13]. Existing approaches for cell annotation are contingent upon unsupervised clustering techniques, necessitating subsequent manual curation and visual validation, a process that can be markedly labor-intensive and requires domain-specific expertise. Achieving higher annotation accuracies can thus be an arduous process due to the iterative steps involved[8]. Therefore, there is a need for scalable computational methods that can accurately classify cells from spatial proteomics data. Promising automated approaches developed recently include probabilistic inferential approaches[14,15], and convolutional neural networks[16,17]. Geuenich et al. introduced ASTIR, an automated method for assigning cell identities using single-cell multiplexed imaging data. ASTIR utilizes a probabilistic model that incorporates prior knowledge of marker proteins to categorize cells into specific cell types. Amitay et al. introduced Cell-Sighter, a cell classification pipeline based on deep learning. This method exhibits promising classification performance. However, it is noted for its shortcomings in computational efficiency due to its reliance on an ensemble of ten ResNet50-based models with random initializations, which can be computationally intensive and potentially resource-consuming.

Therefore, a computationally lightweight and fast automated cell classification method, while achieving human-level accuracy, is required to improve the efficiency and scalability of spatial proteomics data analysis. We present here MAPS (Machine learning for Analysis of Proteomics in Spatial biology), a machine learning package that enables accurate and fast cell annotation with the highest in-class performance when benchmarked across multiple spatial proteomics platforms. MAPS can facilitate both the speed and quality of cell annotation process so that researchers can allocate more downstream efforts in unveiling novel biological processes in situ.

## Results

### Development of MAPS and initial application to an in-house curated cHL MIBI dataset

Herein, we postulated that a feed-forward neural network would be an efficient and robust model for rapid and accurate cell phenotyping. This model, MAPS, predicts the cell class from a set of user-defined classes using the expression of a cell for $N$ markers, and its area in pixels (Fig. 1A). MAPS employs four fully connected hidden layers with ReLU activation function and dropout layers, followed by a classification layer with softmax function. MAPS accurately predicted the cell phenotypes in healthy and diseased tissues, as exemplified by a MIBI dataset of classic Hodgkin Lymphoma (cHL) [1669853 cells, 13 cell types] (Fig. 1B, Supplementary Fig. 1A). All ground truth annotation data was generated through traditional iterative clustering and visual inspection, followed by final inspection by a board certified pathologist (S.J.R.). All questionable clusters were subject to further clustering based on the key markers that were present, and difficult cell types, such as Reed-Sternberg tumor cells in cHL, were then subject to manual annotation to generate the "ground truth" reference (Supplementary Fig. 1B, further expanded in METHODS). We next evaluated the performance of MAPS, including precision, recall, and F1-scores in a stratified 5-fold cross validation (Fig. 1C, D; see METHODS for more details). For each model training, four folds were used for the training/validation set and the remaining fold was used for the test set. The optimal model was chosen based on a validation set. We ensured that data corresponding to a specific case was exclusively part of either the training/validation or the test set. The mean cell expression matrix of various phenotypic markers

for each cell type in the ground truth and predictions had high concordance (Fig. 1E). MAPS thus demonstrated a consistently high accuracy in predicting the cell type from spatial proteomics datasets.

### Benchmarking comparisons of MAPS against other methods and on other spatial proteomics data

We sought next to demonstrate real world practicality of MAPS, and its performance against other state-of-the-art approaches, ASTIR[14] and CellSighter[16]. We collected and annotated in-house data from (1) MIBI on cHL using a first cohort (cHL 1; 1669853 cells), (2) MIBI on cHL using a second cohort (cHL 2; 192795 cells), and (3) CODEX on cHL (145161 cells). MAPS, ASTIR and CellSighter were trained on the same ground truth data generated on the aforementioned datasets in the same manner (see METHODS), and the resulting phenotype maps visualized (Fig. 2A and Supplementary Fig. 2A). To evaluate the performance on a different disease type, we also trained, validated and tested all three models on an external MIBI dataset from diffuse large B cell lymphoma (DLBCL) study[18]. The analysis of the precision, recall and F1 scores indicated the consistently highest performance of MAPS across all datasets, followed by CellSighter and ASTIR (Fig. 2B, Supplementary Figs. 2B, 3, 4, and Supplementary Tables 4–15). All three methods show relatively small performance variance on cHL (CODEX) compared to the other datasets (cHL 1 MIBI, cHL 2 MIBI, and DLBCL MIBI). We postulated that this small performance variance could be attributed to the different data-splitting strategies. The cHL (CODEX) dataset, consisting of a large single tissue image from one patient, was split at the cell level, which can lead to bias and overfitting in the machine learning model. This is because adjacent cells in the same image may have been split between the training, validation, and test sets, potentially leading to high overlap in the distribution of cells in training, validation and test sets. This can artificially result in higher performance in the test set that may not generalize well to new samples. In contrast, the other three datasets consist of multiple regions and patient cases, and were split at the case level, which prevents this issue of information leakage, thus resulting in a more realistic real-world performance. Details of these datasets are further elaborated in the METHODS.

Given the high performance of MAPS and CellSighter, we next computed precision-recall curves and average precision per class to gain further insights on the model differences (Supplementary Figs. 3, 4). MAPS demonstrated consistent performance across all datasets and consistently outperformed CellSighter on all four datasets, with average precision ranging from 0.93 to 0.99 for MAPS, and 0.75 to 0.97 for CellSighter.

The evaluation of MAPS counterparts on in-house datasets may suffer from retraining bias. Therefore, we further benchmarked MAPS on a public data set from a study on colorectal cancer (CRC) generated with CODEX[7] and compared its performance with previously published results of CellSighter (an ensemble of 10 randomly initialized Cell-Sighter models) on same dataset. MAPS continued to perform robustly over CellSighter, also highlighting its compatibility with different tissue types (Fig. 3A).

The generalizability of MAPS was next tested by applying the model trained on cHL1 to predict cell phenotypes of cHL2. Its performance was compared against the cross-dataset performance of ASTIR and CellSighter (Fig. 3B). Overall, all three models showed a decrease in performance, with the performance of MAPS and CellSighter still acceptable for most of the cell types (F1-Scores vary between 0.5–0.6). However, due to the heterogeneity among different datasets and the lack of established baseline for proteomics data, which, in turn, causes difficulties to adjust for the heterogeneity, a decrease in performance was expected.

### Data efficiency and computational resource usage

Since large and well-annotated datasets are not always readily available and require tremendous efforts to produce, we tested

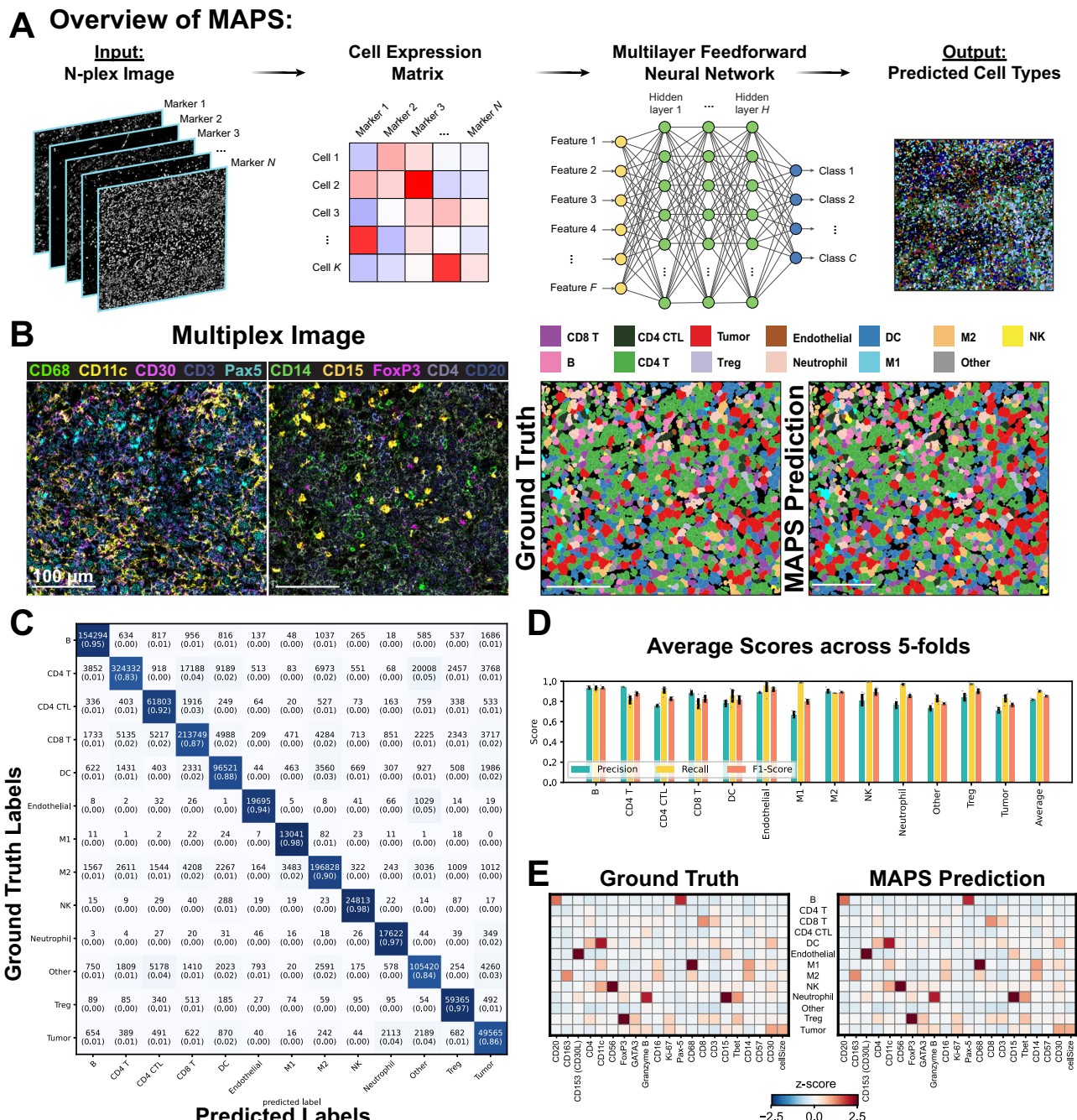

**Fig. 1 | Overview of MAPS architecture and its performance on cHL1 (MIBI) dataset across 5-folds cross validation. A** Schematic of MAPS for machine learning based cell phenotyping. MAPS takes a multiplex image as input and converts it into a cell expression matrix after preprocessing, which is then fed into a feedforward neural network for cell type prediction from a predefined list of classes. **B** A representative FOV of a multiplexed image used for cell phenotyping. Cell phenotype maps generated via manual annotation (Ground Truth) and MAPS (MAPS Prediction) are shown for visual comparison. For each dataset, the imaging was performed once on all FOVs. Source data are provided as a Source Data file. **C** Confusion matrix of MAPS predictions. Numbers in parentheses indicate the percentage of cells with respect to total cells in the corresponding row/class. **D** Average precision, recall, and F1-score of MAPS predictions across five folds. Error bars represent ±1 standard deviation centered around the mean. Source data are provided as a Source Data file. **E** Average cell marker expression matrix for each cell type generated using ground truth labels (left) and MAPS prediction (right).

the performance of MAPS on various differently sized data sets. To do so, we randomly sampled 5%($\bar{n}=48483$), 10%($\bar{n}=96967$), 25%($\bar{n}=242418$), 50%($\bar{n}=484833$), 75%($\bar{n}=727247$), and 100%($\bar{n}=969660$) from the cHL 1 MIBI training set. Overall, MAPS performed comparably well across all of the training sets with, as expected, increasing performance as the size of the training set increases (Fig. 4A). However, a diminishing return was observed as the size of the training set increased, indicating that optimal

performance of MAPS can be achieved with moderately sized data sets, as long as the annotated subsampled cells represent the overall cell type populations.

Given how neural network models can be resource intensive, we next quantified the level of computational resource usage between MAPS and CellSighter. Here, we used the cHL (CODEX) dataset due to its relatively small size yet diverse number of cell type representations. We observed comparable total run time and

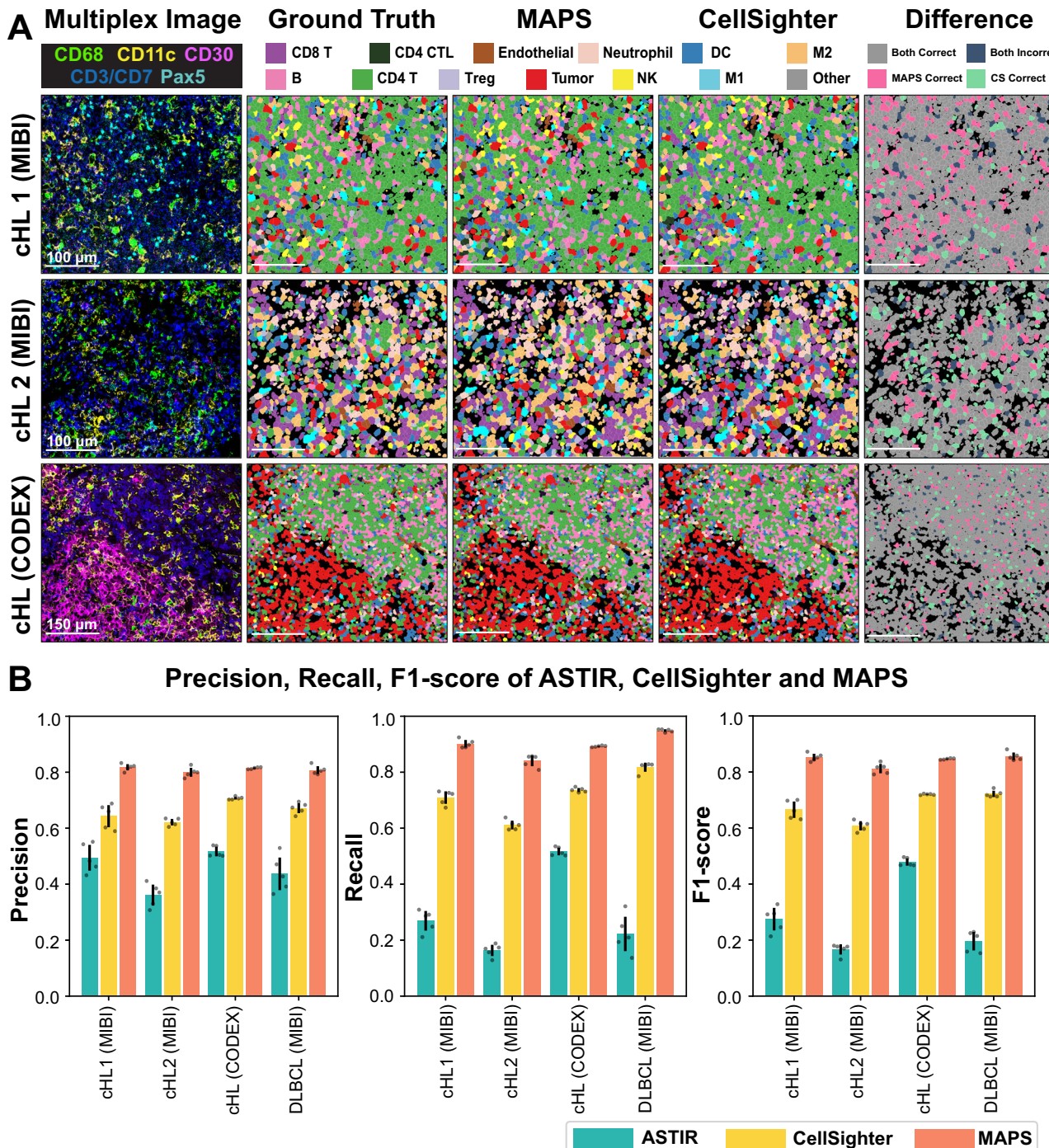

**Fig. 2 | Visual and quantitative comparison of MAPS performance with its counterparts. A** Comparison of MAPS and CellSighter performances across three multiplex image datasets. The last column indicates differences in cell predictions between these two methods. Row 1: Representative cHL FOV from a cHL patient cohort (cHL 1) acquired via the MIBI. Row 2: Representative cHL FOV from another cHL patient cohort (cHL 2) acquired via the MIBI. Row 3: Representative cHL FOV from a separate cHL tissue acquired via the CODEX. For each dataset, the imaging was performed once on all FOVs. Source data are provided as a Source Data file. **B** Comparisons of precision, recall, and F1 score among ASTIR, CellSighter, and MAPS, across three in-house datasets (cHL1 MIBI, cHL2 MIBI, cHL CODEX) and an external dataset DLBCL MIBI. Error bars represent ± 1 standard deviation centered around the mean. For cHL1 (MIBI), $n = 1669853$ cells. For cHL2 (MIBI) $n = 230895$ cells. For cHL (CODEX), $n = 145161$ cells. For DLBCL (MIBI), $n = 338798$ cells. Source data are provided as a Source Data file.

GPU memory utilization between MAPS and ASTIR, with substantially higher values for CellSighter. Memory utilization was similar between MAPS and CellSighter, with lower values for ASTIR (Fig. 4B). Our results highlight the well-balanced computational efficiency and rapid performance of MAPS, relative to its top-in-class accuracy for cell type annotation.

## Discussion

In this study, we introduced MAPS, for pathologist-level accuracy in cell annotation from spatial proteomics data. Our comprehensive evaluation demonstrates that MAPS outperforms its counterparts, ASTIR and CellSighter, in terms of both accuracy and computational efficiency, thus establishing it as a robust tool for precise cell type prediction.

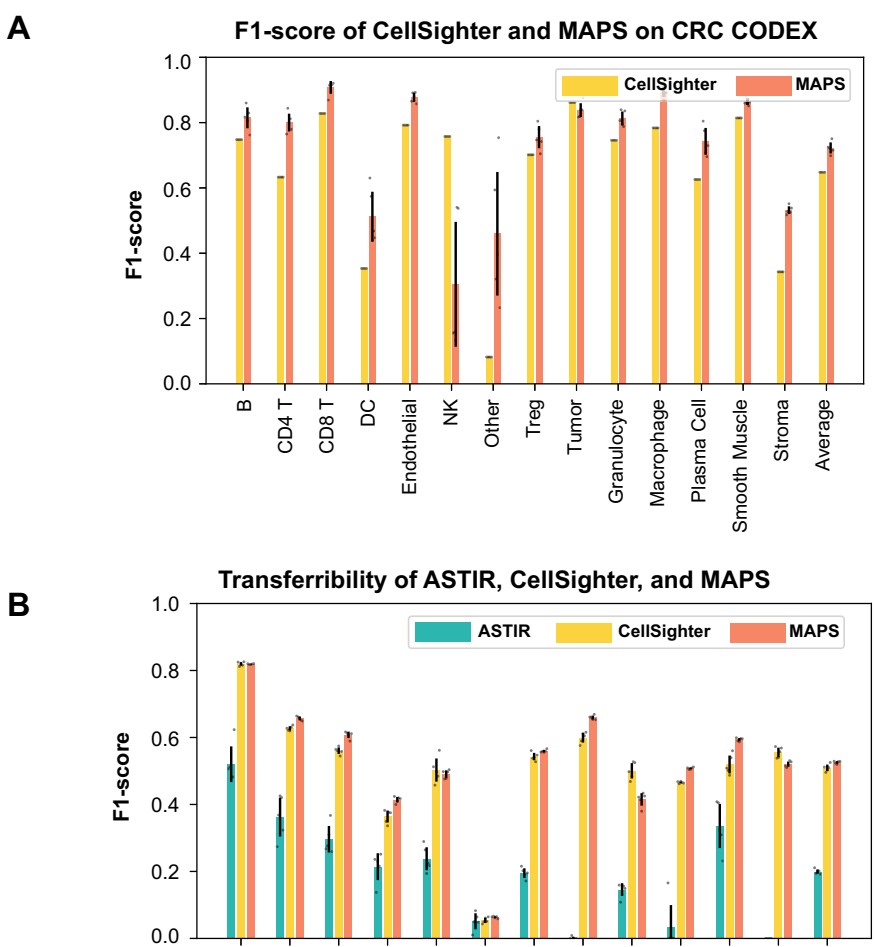

**Fig. 3 | Quantitative comparisons of the performance of MAPS when applied to external datasets and applied across datasets. A** Comparisons of the cell-phenotype-specific F1 scores of MAPS and Cellsighter on the CRC CODEX dataset. CellSighter's results were calculated based on the precision and recall reported in the original publication (Amitay et al. 2022), and, therefore, no measurement of uncertainty was shown. For MAPS, error bars represent ±1 standard deviation. For CRC CODEX, $n = 258385$. Source data are provided as a Source Data file.
**B** Comparisons of the cell-phenotype-specific F1 scores of ASTIR, CellSighter, and MAPS when the respective models were trained on cHL1 MIBI and tested on cHL2 MIBI. Error bars represent ±1 standard deviation centered around the mean. For cHL1 MIBI, $n = 1669853$ cells. For cHL2 MIBI, $n = 230895$ cells. Source data are provided as a Source Data file.

MAPS exhibits superior performance metrics compared to existing state-of-the-art methods. Specifically, it achieves significantly higher F1-scores, precision, and recall, showcasing its remarkable ability to accurately discern cell types from spatial proteomics data (Fig. 2B). This heightened performance is a testament to the effectiveness of the feed-forward neural network architecture employed in MAPS. This architecture enables the efficient processing of spatial proteomics data, allowing for the capture of intricate relationships between input features and cell types. The incorporation of ReLU activation functions introduces non-linearity, further enhancing the model's capacity to discern complex cellular patterns. The integration of dropout layers during training mitigates overfitting, bolstering the model's generalization capabilities.

The strength of MAPS lies in its consistently high performance across diverse biological contexts (Figs. 2B, 3A). It demonstrates proficiency in handling various disease models such as classical Hodgkin lymphoma (cHL), diffuse large B cell lymphoma (DLBCL), and color-ectal cancer (CRC). This adaptability underscores the versatility of MAPS, positioning it as a reliable tool for a wide range of biological and biomedical research applications. Furthermore, MAPS exhibits exceptional cross-platform compatibility, performing consistently well on both MIBI and CODEX datasets (Figs. 2B, 3A). This feature is of paramount importance, as it ensures the applicability of MAPS in diverse experimental settings. In addition, a reasonable level of generalizability across datasets further solidifies the position of MAPS as a leading method for cell annotation in spatial proteomics data (Fig. 3B).

In term of data efficiency, MAPS shows consistent performance when trained with limited training data (Fig. 4A) in addition to its exceptional performance in well-sampled scenarios. This capability enables accurate cell type annotation even in situations where data availability may be constrained. The optimal performance of MAPS can be achieved with moderately sized dataset, as long as the annotated cells represent their respective populations well.

Finally, MAPS not only surpasses its counterparts in terms of accuracy but also stands out for its computational efficiency (Fig. 4B). Its training time is orders of magnitude faster than existing supervised methods, a crucial advantage in the analysis of large-scale spatial proteomics data. This efficiency is a pivotal feature, particularly in scenarios where rapid processing of extensive datasets is imperative. By integrating MAPS into current spatial proteomics workflows, it can

**A**

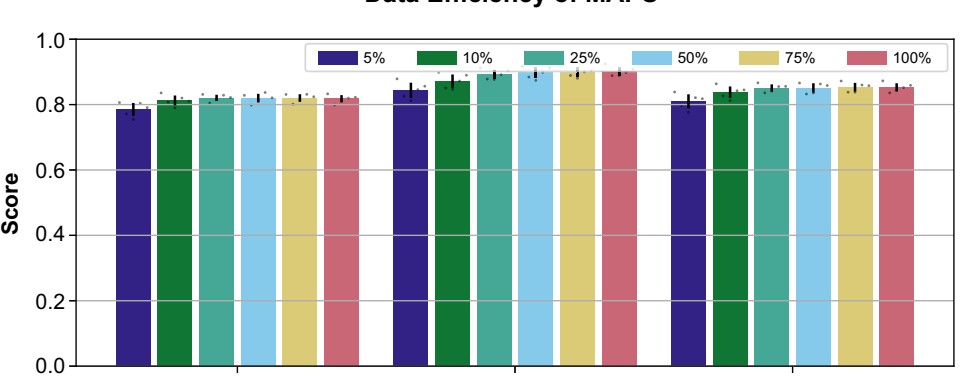

**B**

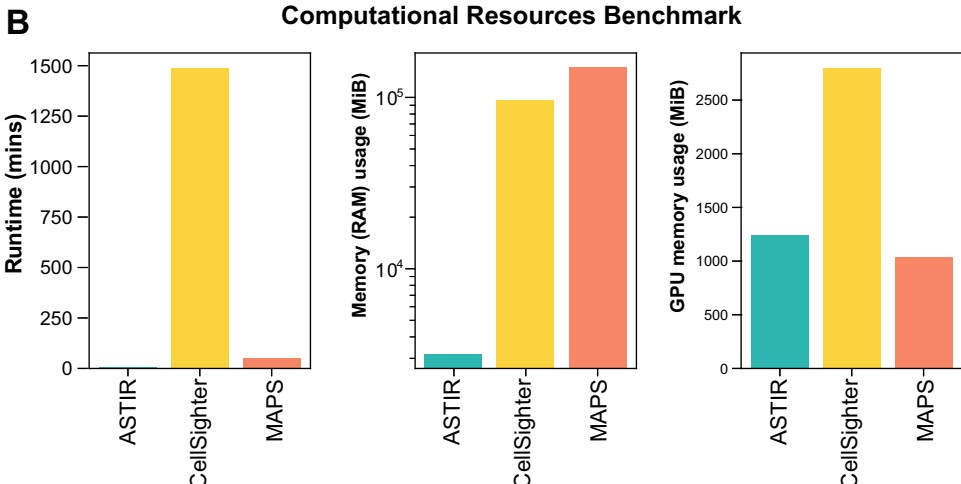

**Fig. 4 | Data efficiency and computational efficiency of MAPS. A** Precision, recall, and F1 scores of MAPS when trained with 5%($\bar{n}$ = 48483), 10%($\bar{n}$ = 96967), 25%($\bar{n}$ = 242418), 50%($\bar{n}$ = 484833), 75%($\bar{n}$ = 727247), and 100%($\bar{n}$ = 969660) of the cHL1 training dataset and tested on the cHL1 testing set. Error bars represent ± 1 standard deviation centered around the mean. Source data are provided as a Source Data file. **B** Comparisons of the run time, memory usage, and GPU memory usage of ASTIR, CellSighter, and MAPS.

expedite the annotation process from smaller, curated "ground truth" datasets, exemplifying its potential to streamline research endeavors in the field (Supplementary Fig. 1B).

In conclusion, the combination of superior performance, simple model architecture, fast training and inference, cross-platform compatibility, and adaptability to different tissue types and disease models firmly establishes MAPS as a powerful tool for cell annotation in spatial proteomics data. The release of the MAPS package and associated data resources on GitHub (https://github.com/mahmoodlab/MAPS) marks a significant contribution to the scientific community, providing researchers with a valuable resource to advance the field of tissue spatial-omics and accelerate discoveries in cellular biology across diverse biological contexts.

## Methods
### Section 1: dataset acquisition
**Ethical statement.** Formalin-fixed paraffin-embedded (FFPE) excisional biopsies from 23 patients with newly diagnosed cHL, and one reactive lymph node were retrieved from the archives of Brigham and Women's Hospital (Boston, MA) with institutional review board approval (IRB# 2010P002736) and patient wavier of consent. Sex and

gender were not considered in the study design due to the proof-of-concept nature of this methodological study. All tumor regions were annotated by V.S. and S.J.R.

**Antibody conjugation and panel.** Lanthanides conjugated antibodies for MIBI were acquired as previously described[19] using the Maxpar X8 Multimetal Labeling Kit (Fluidigm, 201300) and Ionpath Conjugation Kits (Ionpath, 600XXX) with slight modifications to manufacturer protocols. In short, 100 µg BSA-free antibody was first washed with the conjugation buffer, then reduced using 4 $mu$mol L$^{-1}$ (final concentration) of TCEP (Thermo Fisher Scientific, 77720) to reduce the thiol groups for 30 min in a 37 °C water bath. The reduced antibody was mixed and incubated with Lanthanide-loaded polymers for 90 min in a 37 °C water bath, then washed for 5 times with an Amicon Ultra filter (Millipore Sigma, UFC505096). Resulting conjugated antibodies were then buffered with at least 30% v/v Candor Antibody Stabilizer (Thermo Fisher Scientific, NC0414486) including 0.02% w/v of sodium azide, and stored at 4 °C until usage.

Oligo conjugation to antibodies for CODEX was performed as previously described[13]. In short, 100 µg BSA-free antibody was reduced using 2.5 mmol L$^{-1}$ of TCEP at RT for 30 min to reduce the thiol groups.

Maleimide-labeled oligos are resuspended in High-salt Buffer C (1 mol $L^{-1}$ NaCl) and incubated with the reduced antibodies at RT for 2 h. The resulting conjugated antibodies are then washed for 3 times in high salt PBS (0.9 mol $L^{-1}$ NaCl) in a 50 kDa centrifugal column (Sigma, UFC505096), buffered with at least 30% v/v Candor Antibody Stabilizer (Thermo Fisher Scientific, NC0414486) supplemented with 0.02% w/v of sodium azide, and stored at 4 °C.

The antibody panels can be found in Supplementary Table 1.

**Gold slide preparation.** The protocol of preparing gold slides has been described previously[5,6,20]. In short, Superfrost Plus glass slides (Thermo Fisher Scientific, 12-550-15) were first soaked and briefly supersonicated in a ddH$_2$O diluted with dish detergent, cleaned by using Microfiber Cleaning Cloths (Care Touch, BD11945) then rinsed in flowing water to remove any remaining detergent. After that, the slides were air-dried with a constant stream of air in the fume hood. The coating of 30 nm of Tantalum followed by 100 nm of Gold was performed by the Microfab Shop of Stanford Nano Shared Facility (SNSF) and New Wave Thin Films (Newark, CA).

**Coverslip and slides vectabonding.** To introduce positive charges for better adhesion of tissue sections onto the surface, pre-cleaned 22x22 mm glass coverslips (VWR, 48366-067) or the e-beam coated gold slides were silanized by VECTABOND Reagent (Vector Labs, SP-1800-7) per the protocol from the manufacturer. The slides were first soaked in neat acetone for 5 min, then transferred into 1:50 diluted VECTABOND Reagent in acetone and incubated for 10 min. After that, slides were quickly dipped in ddH$_2$O to quench and remove remaining reagents, then tapped on Kimwipe to remove remaining water, the resulting slides were air-dried then stored at room temperature.

**MIBI retrieval and staining protocol.** The procedure of a general MIBI staining is similar to previously described[5,8,21]. The FFPE block was sectioned onto Vectabond-treated gold slides by 5 μm thickness. The sections then went through a standard deparaffinization and antigen retrieval process. In brief, slides with FFPE sections were first baked in an oven (VWR, 10055-006) for 1 h at 70 °C, then were transferred into neat xylene and incubated for 2x 10 min. Standard deparaffinization was performed with a linear stainer (Leica Biosystems, ST4020) in the following sequence: 3x neat xylene, 3x 100% EtOH, 2x 95% EtOH, 1x 80% EtOH, 1x 70% EtOH, 3x ddH$_2$O, 180 s each step with constant dipping, then rest in ddH$_2$O. Antigen retrieval was then performed at 97 °C for 10 min with Target Retrieval Solution (Agilent, S236784-2) on a PT Module (Thermo Fisher Scientific, A80400012).

After PT Module processing, the cassette with slides and solution was left on the benchtop until it reached room temperature. After a quick 1x PBS rinse for 5 min, the sections were blocked by BBDG (5% NDS, 0.05% sodium azide in 1x TBS IHC wash buffer with Tween 20), then stained at 4 °C in an antibody cocktail for overnight (Supplementary Table 1). Subsequently, the samples were quickly rinsed with 1x PBS, then fixed by the Post-fixation buffer (4% PFA + 2% GA in 1x PBS buffer) for 10 min, then quenched with 100 mM Tris HCl pH 7.5, before undergoing a series of dehydration steps on the linear stainer (3x 100 mM Tris pH 7.5, 3x ddH$_2$O, 1x 70% EtOH, 1x 80% EtOH, 2x 95% EtOH, 3x 100% EtOH, 60 s for each step), before store in a vacuum desiccator until acquisition.

**CODEX retrieval and staining protocol.** The procedure for CODEX staining is similar to previously described[22]. A cHL FFPE section was mounted on a No.1 glass coverslip pre-treated with VECTABOND Reagent (Vector laboratories, SP-1800-7) as described above, and deparaffinized by heating at 70 °C for 1 h, followed by two 15-min soaks in a xylene bath. The tissue was then manually rehydrated in 6-well plates by incubating in 2x 100% EtOH, 2x 95% EtOH, 1x 80% EtOH, 1x 70% EtOH, and 3x ddH$_2$O, for 3 min each with gentle rocking. Heat-induced antigen retrieval (HIER) was performed in a coverslip jar containing 1x Dako pH 9 Antigen Retrieval Buffer (Agilent, S2375) while using a PT module filled with 1x PBS; the PT module was set to pre-warm to 75 °C, heat to 97 °C for 20 min, before cooling to 65 °C. After HIER, the tissue was washed in CODEX hydration buffer (Akoya Biosciences, 232105) 2x for 2 min and incubated in CODEX staining buffer (Akoya Biosciences, 232106) for 20 min. The tissue was then transferred to a humidity chamber to block with 200 μL of BBDG while being photobleached with a custom LED array for 2 h (see below), then stained at 4 °C in an antibody cocktail overnight.

The blocking buffer was prepared by combining 180 μL of BBDG block, 10 μL of oligo block, and 10 μL of sheared salmon sperm DNA. The BBDG block was prepared by mixing 5% donkey serum, 0.1% Triton X-100, and 0.05% sodium azide in 1x TBS IHC Wash buffer with Tween 20 (Cell Marque, 935B-09). The oligo block was prepared by mixing 57 different custom oligos (IDT) to create a master mix with a final concentration of 0.5 $\mu$mol $L^{-1}$ per oligo. The sheared salmon sperm DNA was used directly from its original 10 mg/mL stock (ThermoFisher, AM9680). To create a humidity chamber, an empty pipette tip box was filled with ddH$_2$O and wet paper towels and then placed on top of a cool box (Corning, 432021) containing an ice block. Two happy lights (Best Buy, 6460231) were leaned against either side of the humidity chamber, and an LED grow light (Amazon, B07C68N7PC) was positioned above. Staining antibodies (Supplementary Table 2) were prepared while blocking.

After overnight antibody staining, the tissue was washed 2x in CODEX staining buffer for 2 min each. Subsequently, it was fixed with 1.6% paraformaldehyde (PFA) with gentle rocking for 10 min; the PFA solution was made by diluting 16% PFA with CODEX storage buffer (Akoya Biosciences, 232107). The tissue was then washed 3x in 1x PBS, incubated in cold 100% methanol for 5 min on ice, and washed 3x with 1x PBS again. All steps except the methanol incubation were performed in 6 well plates with gentle rocking. The tissue was then fixed with CODEX final fixative for 20 min at RT in a humidity chamber; the final fixative was prepared by mixing 20 $\mu$L of CODEX final fixative (Akoya Biosciences, 232112) in 1000 μL of 1x PBS. Finally, the tissue was rinsed 3x in 1x PBS and stored in 1x PBS at 4° until CODEX image acquisition.

**MIBI-TOF imaging.** Datasets were acquired on a commercially available MIBIscope$^{TM}$ System from Ionpath (Production) equipped with a Xenon ion source (Hyperion, Oregon Physics). The typical running parameters on instruments are listed as following:

- Pixel dwell time: 2 ms
- Pixel dwell time: 2 ms
- Image area: 400 × 400 μm
- Image size: 512 × 512 pixels
- Probe size: 400 nm
- Primary ion current: 4.9 nA on a builtin Faraday cup (or the "Fine" imaging mode)
- Number of depths: 1 depth

After acquisition, images were extracted with the toffy package (toffy notebook 3b). Detailed pre-processing is mentioned in the sections below.

**CODEX imaging.** A black flat bottom 96-well plate (Corning, 07-200-762) was used for the reporter plate, where each well represented an imaging cycle. Each well was filled with 240 μL of plate master mix, containing DAPI nuclear stain (7000003, Akoya) (1:600) and CODEX assay reagent (Akoya Biosciences, 7000002) (0.5 mg/mL), as well as two fluorescent oligonucleotides (5 μL each) on the Cy3 and Cy5 channels. Blank channels were also included in the first and last wells, with plate master mix substituted for fluorescent oligonucleotides. The plate was then sealed with aluminum film and stored at 4 °C until CODEX image acquisition.

Prior to CODEX image acquisition, the tissue coverslip and reporter plate were placed into the CODEX microfluidics instrument. The coverslip was stained with 750 μL nuclear stain solution for 3 min before being washed by the fluidics device; the nuclear stain solution was prepared by mixing 1 μL of DAPI nuclear stain in 1500 μL of 1x CODEX buffer. CODEX imaging was operated under a 20x/0.75 objective (CFI Plan Apo $\lambda$, Nikon) mounted to an inverted fluorescence microscope (Keyence, BZ-X810) connected to the CODEX microfluidics instrument and CODEX driver software, and the DAPI stain was used to set up imaging areas and z planes. Each imaging cycle contained three channels - DAPI, Cy3, Cy5—and images taken on the first and last cycles were used as blanks for background correction. Multiplexed images were stitched and background corrected using the Singer software (v1.0.7) from Akoya.

### Section 2: dataset pre-processing

**Channel crosstalk removal.** Similar to fluorescence imaging, mass-spectrometry imaging such as MIBI also has channel crosstalk due to the formation of adducts[6] or isotopic impurity of the elemental labels used. Thus, Rosetta algorithm was applied to extracted raw images to remove noise from channel crosstalk in a manner similar to flow-cytometry data (toffy notebook 4a). In addition to that, as background signals from bare slides and organic fragments can be partially reflected by gold and "Noodle" background channels, those counts were also removed with a fine-tuned coefficient matrix along with channel crosstalk. This step was performed with a local implementation of toffy package with minor modification.

**Image denoising.** Image noise in multiplex images is a well-known issue caused by various factors such as instrumentation, tissue quality, and non-specific binding of antibodies. To tackle this challenge, a deep learning-based method is proposed that poses image denoising as a background-foreground segmentation problem. In this approach, the real signal is considered as foreground, while the noise is considered as background. The proposed method uses a supervised deep learning-based segmentation model, UNET[23], to segment the foreground from the given image. To train the model, ground truth is generated using a semi-supervised kNN-based clustering method[24]. The kNN-based clustering method helps to generate reliable ground truth for the model training. Once the model is trained, it is applied to all markers in all images to obtain predicted foreground segmentation maps. These segmentation maps are then multiplied with the original images to get rid of noise and obtain clean images.

**Cell segmentation.** Cell segmentation of the MIBI cHL datasets was performed with a local implementation of deepcell-tf 0.6.0 as described[11,25]. Histone H3 channel was used for the nucleus, while the summation of HLA-DR, HLA1, Na-K-ATPase, CD45RA, CD11c, CD3, CD20, and CD68 was used as the membrane feature. Signals from these channels were first capped at the 99.7th percentile before input into the model.

Cell segmentation of the CODEX cHL dataset was performed using a local implementation of deepcell-tf 0.12.2. Segmentation was done using DAPI as the nuclear channel and a summation of CD4, CD7, CD15, CD30, CD11b, CD20, CD45RA, CD45RO, CD31, Podoplanin, and HLA-DR as the membrane features to ensure ideal segmentation of all cell types in the singular field of view.

The deepcell-tf version used to generate the final segmentation mask, along with the detailed parameters for cell segmentation are summarized in Supplementary Table 3.

**Image intensity normalization.** Due to instrumental limitation, the FOV that MIBI routinely acquired is only 400 × 400 μm size, stitching to achieve large tissue acquisition, and thus the across FOV difference is unavoidable. To compensate for the inter FOV difference, a set of scripts were developed and integrated into the data processing pipeline. Briefly, in a stitched run, the average Histone H3 counts under cell segmentation masks of each FOV were calculated, then, all FOVs Histone H3 counts were normalized towards the highest counts, while other channels were multiplied by the same coefficient. Additional flattening based on the Histone H3 counts were also used to avoid boundary effects and image biases. The code and parameters used are available in the analysis pipeline section.

**Image to cell expression matrix and across-runs normalization.** The counts of each channel inside each cell segmented mask were summed up and then divided by the cell size to create the cell expression matrix based on normalized stitched TIFs along with their segmentation mask. To avoid the across-runs derivation, the median value of per cell Histone H3 of each run was calculated, then all runs medians of Histone H3, along with all other channels counts were normalized towards the highest Histone H3 median value of that MIBI dataset. The code and parameters used are available in the analysis pipeline section.

**Generation of cell phenotyping ground truth.** Cell phenotyping on the cHL MIBI datasets was accomplished through an iterative clustering and annotating process. The clustering was performed with FlowSOM[26] on the cHL 1 dataset and Leiden[27] on the cHL 2 dataset. The cHL 1 dataset was initially clustered with CD11c, CD14, CD15, CD153, CD16, CD163, CD20, CD3, CD30, CD4, CD56, CD57, CD68, CD8, FoxP3, GATA3, Granzyme B, and Pax-5 to capture most of the cell types present in the data. The resulting clusters were then manually annotated by examining the predominantly enriched markers of each cluster, which was done by plotting Z-score and mean expression heatmaps across all clusters and the phenotypic markers used. Clusters with a clear enrichment pattern were annotated. Next, with Mantis Viewer[28], the assigned annotation was confirmed by mapping the annotation to each cell and overlaying the raw images of the enriched markers for visual inspection. Due to noise in the data, there were certain clusters with unclear enrichment patterns. These clusters were assessed based on the phenotype marker enrichment patterns and subjected to further clustering and visual inspection. This interactive process was repeated until no useful information could be further extracted, and the remaining cells with no clear enrichment pattern were assigned as "Others". For the cHL 1 dataset, 1538433 out of 1669853 cells (92.2%) were assigned a final annotation.

Cell phenotyping on the cHL CODEX dataset was performed through an iterative process using Rphenoannoy (R implementation of PhenoGraph) and FlowSOM[26,27] to cluster on CD30, CD20, CD2, CD7, CD8, CD57, CD4, Granzyme B, CD56, FoxP3, CD11c, CD16, CD206, CD163, CD68, CD15, CD11b, Cytokeratin, Podoplanin, CD31, MCT, and a-SMA. The resulting stratified cell clusters and corresponding enriched phenotypic markers were then visualized with Z-score and mean expression heatmaps. Cells were then individually mapped back to the original tissue images in QuPath 0.2.0-m1 to validate marker enrichment. Clusters with clear enrichment patterns for a particular cell type were annotated accordingly. Clusters with unclear or partially correct enrichment patterns were further clustered using FlowSOM based on a curated subset of phenotypic markers present on these unclear populations. Multiple iterations of clustering and annotation were performed until signal-noise ratio was too low to confidently distinguish the phenotype of the remaining cells, which were assigned as "Others". 140,053 out of 145,161 cells (96.5%) were assigned a final annotation.

All final annotations were assessed by S.J. and S.J.R (a board certified hematopathologist).

### Section 3: datasets overview
Our study utilized five different datasets, including three in-house datasets, for cell phenotyping in cHL, DLBCL, and CRC. The cHL 1, cHL

2, and DLBCL[18] datasets were acquired using Multiplexed Ion Beam Imaging (MIBI) and contained cells from 13, 12, and 9 different phenotypes, respectively. The cHL CODEX and CRC CODEX[7] datasets were acquired using Co-detection by Indexing (CODEX) and contained cells from 16 and 14 different phenotypes, respectively. The datasets had varying numbers of cells, protein/functional markers, and levels of class imbalance, and were splitted into five-folds where four folds were used as training/validation (80%/20%) sets and the remaining fold was used as the test set, iteratively.

**cHL 1 and cHL 2 (MIBI) dataset.** The cHL 1 and cHL 2 (MIBI) Datasets are two in-house datasets used in our study for cell phenotyping in cHL. Both sets of samples were stained with the same batch of antibody cocktail (Supplementary Table 1) with 46 protein/functional markers, and acquired using Multiplexed Ion Beam Imaging (MIBI). cHL 1 Dataset contains 166,9853 cells from 18 cHL patients and 1 control rLN, while cHL 2 Dataset has over 230,895 cells from six FOVs—five from cHL patients and one from a control rLN. When training the proposed method, 5 markers from the cHL 1 dataset were dropped due to poor staining quality, while all 46 markers remained in the training set of cHL 2. To evaluate the performance of our proposed method, both datasets were split into 5 folds for multi-fold training and testing of the proposed method, and under both cases, the FOVs of the control cases were part of the training set in each fold.

**cHL (CODEX) dataset.** The cHL (CODEX) dataset is another in-house dataset that was acquired using Co-Detection by Indexing (CODEX), a multiplex imaging technique that allows for simultaneous detection of over 50 markers. The dataset consists of a single large FOV containing over 145,161 cells. The cells in the cHL (CODEX) dataset are classified into 16 different cell phenotypes, and each class has an average of 8000+ cells. The multiplex FOV in this dataset consists of 49 markers, which include different markers than those used in the cHL 1 (MIBI) and cHL 2 (MIBI) datasets (see Supplementary Table 2 for more details). To evaluate the performance of MAPS, we randomly split the cells in the cHL (CODEX) dataset into five folds using stratified sampling to ensure a balanced number of cells in each fold for each class.

**CRC CODEX dataset.** The CRC CODEX dataset (DOI: 10.17632/mpjzbtfgfr.1) is a public dataset that we used in our study to evaluate our proposed method for cell phenotyping[7]. It consists of 258,385 cells from 14 different classes, with a large variation in the number of cells per class, ranging from as low as 323 cells to as high as >47,000 cells. For our study, we used the same markers and classes as described in the CellSighter paper to ensure a fair head-to-head comparison with MAPS. Since there was no information available about the training and validation split in the dataset, we adopted the five-fold cross-validation approach.

**DLBCL MIBI dataset.** The DLBCL MIBI dataset used in this paper is from a previous publication with participation of some of coauthors[18]. It consists of 338,798 cells from 143 FOVs of DLBCL TMA cores of 51 patients, along with 8 FOVs from reactive lymph nodes. In the previous study, those cells were clustered into 9 types by the 10 lineage-associated markers out of the 22-plex image deck. To evaluate the performance of our proposed method, the dataset was split into 5 folds for multi-fold training and testing where FOVs of the control cases were part of the training set in each fold.

## Section 4: MAPS model, training and evaluation
**Model architecture.** The proposed cell phenotyping method used a feed-forward neural network to predict the cell class from a set of predefined classes ($K$). Let $x \in \mathbb{R}^{N+1}$ be the input data, which consists of the expression of a cell for $N$ markers and its area in pixels. The neural network processes this input data to generate a predicted cell

class $y$. The neural network used in the proposed method consists of four fully connected hidden layers, denoted by $h_1$, $h_2$, $h_3$, and $h_4$. Each hidden layer is followed by a ReLU activation function and a dropout layer, denoted by $g_1$, $g_2$, $g_3$, and $g_4$. The output of the last hidden layer, $h_4$, is fed into the classification layer, which generates the predicted cell class $y$. The classification layer uses a softmax function to convert the output of the neural network into a probability distribution over the predefined classes. Let $W_i$ and $b_i$ denote the weights and biases of the $i^{th}$ layer of the neural network, respectively. Then the output $h_i$ of the $i^{th}$ hidden layer can be written as:

$$h_i = g_i(W_i h_{i-1} + b_i) \tag{1}$$

where $h_{i-1} \in \mathbb{R}^{512}$ is the output of the $(i-1)^{th}$ hidden layer or the input $x$ for $i = 1$, and $g_i$ is the activation function for the $i^{th}$ layer, which is the ReLU function in this case. The dropout layers are not included in this equation, as they only modify the output of the hidden layers during training, and do not affect the final output of the neural network. The classification layer computes the predicted cell class $y$ as follows:

$$y = \underset{k}{\arg\max} \ \text{softmax}(W_c h_4 + b_c)_k \tag{2}$$

where $W_c$ and $b_c$ are the weights and biases of the classification layer, and softmax is the softmax function that converts the $k^{th}$ output into a probability distribution over the predefined classes ($K$). The predicted cell class $y$ is the class with the highest probability.

**Training details.** For the training of the proposed method, a batch size of 128 and a dropout probability of 0.25 were used for all datasets. The number of training epochs varied for each dataset due to the varying sizes of the datasets. For larger datasets (cHL 1 MIBI), the number of epochs is set lower, as more training steps are performed within each epoch. Conversely, for smaller datasets (cHL 2 MIBI and cHL CODEX), we utilize a higher number of epochs to ensure an adequate number of training steps. Specifically, the model was trained for 100 epochs on the cHL 1 (MIBI) dataset, and for 500 epochs on all other datasets. Additionally, we implement two essential hyperparameters, namely minimum epoch and patience, specifically designed to address the issue of overfitting. The minimum epoch ensures that the model undergoes a minimum number of training epochs, and the patience parameter enables early stopping if the validation performance does not improve, thus mitigating the risk of overfitting. The model with lowest validation loss was selected as the best model for evaluation on test sets and inference on new data.

## Section 5: evaluation across methods
To evaluate the performance of the proposed method, we employed several evaluation methods. Firstly, we used the confusion matrix to visualize the performance of the model. The confusion matrix displays the number of true positive, false positive, true negative, and false negative predictions made by the model. From the confusion matrix, we calculated the precision, recall, and F1-score metrics. Precision measures the proportion of true positive predictions made by the model out of all the positive predictions made, while recall measures the proportion of true positive predictions made out of all the actual positive instances in the dataset. The F1-score is the harmonic mean of precision and recall and is a balanced measure of both metrics.

Additionally, we used the average precision metric, which measures the area under the precision-recall curve. This metric is particularly useful for imbalanced datasets, where there are more negative instances than positive ones. The average precision metric takes into account the precision and recall values at various thresholds and provides a summary of the model's overall performance.

Finally, we also used the mean cell expression matrix to visualize the expression levels of different markers in the different cell types predicted by the model. This matrix provides a summary of the mean expression levels of each marker in each cell type and can help to identify differences in marker expression between different cell types when compared with the cell expression matrix generated using ground truth labels.

**Comparisons with other methods.** We compared our proposed method with two existing cell phenotyping methods, namely ASTIR and CellSighter. The code for both ASTIR and CellSighter methods is publicly available for reproducibility and comparison purposes.

**ASTIR.** ASTIR is a probabilistic model for cell phenotyping that uses deep recognition neural networks to predict cell types without requiring labels for each cell[14]. Instead, ASTIR only requires a list of protein markers for each expected cell type within a dataset. The method is based on the assumption that each cell type can be characterized by a unique combination of protein markers, and that the expression levels of these markers can be used to classify cells into their respective types. We reported results of the ASTIR method on three in-house datasets. For each dataset, our experts defined the list of protein markers for each cell type. We evaluated the results using five-fold cross-validation, using exactly the same folds as in the proposed method, for a fair head-to-head comparison.

**CellSighter.** The CellSighter is a deep learning based supervised cell classification method[16]. Unlike ASTIR and the proposed method which works on cell expression matrices, CellSighter takes image, cell segmentation mask, and cell to class mapping as input. To evaluate the performance of CellSighter, we re-trained it on the same three in-house datasets using the same 5-fold cross validation splits as in the proposed method. This ensures a fair comparison between the methods. We obtained the CellSighter results on the publicly available CRC CODEX dataset from the paper to avoid any re-training bias while comparing it with the MAPS results on the same dataset.

**Computation resource evaluation across methods.** To evaluate the computation resource usage of each method, we ran the three methods on a Linux platform (2x Intel Xeon 6334 "Ice Lake-SP" 3.6 GHz 8-core 10 nm CPUs; 4x NVIDIA "Ampere" RTX A5000 PCI-E+NVLink 24GB ECC GPU Accelerator/Graphics Cards; 1TB DDR4 memory @ 3200 MHz) using the cHL (CODEX) dataset. During model training and cell type inference of each method, we tracked their CPU, GPU, and memory (RAM) usage using "top", "ps -ef", and "nvidia-smi" commands. For the parallel methods, we recorded the resource usage of all its processes and multiplied it by the number of cores used in parallel.

**Statistics and reproducibility.** We employed data from every accessible sample within each dataset and no statistical method was used to predetermine sample size. Furthermore, in our study design, we incorporated all available markers from in-house datasets, selectively choosing markers from public datasets to harmonize with prior research. Similarly, we restricted our analysis to cell types utilized in previous studies for public datasets. However, for in-house datasets, we only excluded cell types with insufficient sample count. For each dataset, we randomly partitioned the data into five folds for cross-validation. The code, featuring a fixed seed value, is provided to ensure the reproducibility of our results.

**Data visualization.** Single channel and multi-color images were assembled and visually inspected with either ImageJ[29], Qupath[30], and Mantis Viewer[28]. Visualizations of the analysis results were either produced using Excel, or R packages "ggplot2" and "pheatmap".

## Reporting summary

Further information on research design is available in the Nature Portfolio Reporting Summary linked to this article.

## Data availability

All the data described in this work and Source Data, including channel images and segmentation masks can be accessed at the Zenodo under accession code DOI: 10.5281/zenodo.10067009[31].

## Code availability

The code for prediction and data visualization can be downloaded at https://github.com/mahmoodlab/MAPS[32].

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

## Acknowledgements

The authors thank members of the Jiang and Mahmood laboratories for their discussions. S.J. is supported by NIH DP2AI171139, P01AI177687, R01AI149672, a Gilead's Research Scholars Program in Hematologic Malignancies, and the Bill & Melinda Gates Foundation INV-002704. F.M. and M.S. are funded by R35GM138216, and the Fredrick National Laboratory. G.P.N. is supported by the Rachford and Carlota A. Harris Endowed Professorship. M.A.S and S.J.R. are supported by a Blood Cancer Discoveries Grant Program from the Leukemia Lymphoma Society, The Mark Foundation, and The Paul G. Allen Frontiers Group. This article reflects the views of the authors and should not be construed as representing the views or policies of the institutions that provided funding.

## Author contributions

S.J., F.M., and S.J.R. conceived of the presented idea and planned the experiments. Y.B., Y.Y.Y., H.C., B.Z., and S.J. performed the experiments. M.S., H.Q., and Y.B. performed data analysis and developed the software. V.S. and S.J.R. performed the pathological annotations. M.S., H.Q., Y.B., S.M., J.Y., Y.Y.Y., and J.L.W. participated in the formal analysis of the datasets. M.S. and H.Q., visualized the data. M.S., H.Q., Y.B., and S.J. wrote the manuscript., and S.J., F.M., S.J.R., M.A.S., and G.P.N. supervised the project. S.M., Y.B., and H.Q. contributed equally and have the right to list their name first in their CV. All authors reviewed and edited the final manuscript.

## Competing interests

S.J.R. has received research support from Affimed, Merck, and Bristol-Myers Squibb (BMS), he is on the Scientific Advisory Board for Immunitas Therapeutics and part of Bristol Myers Squibb International Immuno-Oncology Network (II-ON). M.A.S. has research funding from BMS, Bayer, Abbvie, and AstraZeneca, and is on advisory boards for AstraZeneca and BMS. G.P.N. is co-founder and stockholder of IonPath Inc, which manufactures the instrument used in this manuscript, is a co-founder and stockholder of Akoya Biosciences, Inc. and inventor on patent US9909167, and is a Scientific Advisory Board member for Akoya Biosciences, Inc. S.J. has received research funding from Roche, Gilead, and Sanofi not related to this work. The other authors declare no competing interests.
