## [Peer Review File · Nature Communications]

REVIEWERS' COMMENTS:

Reviewer #1 (Remarks to the Author)

MAPS is a neural network-based algorithm trained on "ground-truth" annotated samples (samples annotated by clustering and marker-based, pathologist-confirmed adjudication) that exhibits an impressive improvement on existing algorithms for cell type annotation of spatial data. Some questions raised by my reading that could strengthen or clarify the manuscript:

Training MAPS:

- Could cyTOF data be used crossfunctionally to train MAPS for MIBI classification within a given tissue type?
- Could cell annotation from databases or user-inputted cell protein markers be used to train MAPS?

Using MAPS:

- How will the algorithm be trained by users for new tissues (ie, not lymphoma or LN samples)? Will a training set be provided, or do training, ground-truth data need to be created de novo? On this note, when a reliable reference/training dataset is not available, could MAPS as an alternative simply output a graphical cell clustering with protein markers for each cluster instead of annotation? Related to this, could the algorithm provide as output the markers that define each cell type?
- Some cells are "unassigned"; could an "unassigned" threshold be set by the user, variably depending on the degree of confidence desired?

Misc:

- Were the reference (training) and test images the same? Presumably not, but important to clarify the details here.
- Figure 2A legend -> why are there 2 colors for "Both correct"?

Reviewer #2 (Remarks to the Author)

In the manuscript entitled "MAPS: Pathologist-level cell type annotation from tissue images through machine learning" by Muhammad Shaban et al. authors describe Machine learning for Analysis of Proteomics in Spatial biology (MAPS), a new machine learning approach for pathologist-level accuracy in cell annotation from spatial proteomics data, facilitating the rapid and precise cell type identification.

This work demonstrates that the proposed method outperforms existing state-of-the-art methods in terms of both accuracy and computational efficiency while showing cross-platform compatibility (MIBI and the novel CODEX).

The proposed method holds significant potential to expedite advances in tissue biology and the understanding complex biological systems i.e. in the context of solid tumors.

This work is innovative, the data interpretation and the solidity of novel method are well supported. I believe the work could meet the standards of Nature Communications after addressing the following revisions:

1. I suggest the author consider removing the acronym from the title
2. The font size for the cell type legend in Supplementary Figure 1A and S2B could be increased
3. The introduction section is poor; no explanation is given to MIBI and CODEX, and no state-of-the-art context is provided. Nature Communications is not a sector journal, and due to the high level of the journal, much more attention to details should be provided in guiding also non-

specialized readers. The authors are requested to make an extra effort in shaping the context of this innovative work in relation to the two platforms used and the state of the art of other approaches.

4. By rereading the discussion and conclusion, there is a lack of ability to go in-depth on the innovative points of the work contextualizing in relation to the selected models, to highlight the value of the work better, I suggest increasing a more detailed and point-specific discussion.

Reviewer #3 (Remarks to the Author)

*** Summary:**

The authors present MAPS, a Python package for cell type classification using a feed-forward neural network. They demonstrate the ability to discern over a dozen cell types from in-house datasets of canonical Hodgkin's Lymphoma using a feature set of 49 expression markers and a 50th segmented cell size feature, with high class precision rates ranging from 0.83 to 0.98. MAPS has the advantages of being faster and more accurate than previously published cell type classification tools such as CellSighter and Astir, and the code is open-source with an easy-to-follow vignette on the authors' Github page.

*** Primary concern:**

It is of this reviewer's opinion that the significance of this work is unclear, for the following reasons.

While the FNN itself runs faster than its competitors, the authors' training data consists of more than 1 million cells. Even with the authors' 'cluster and interpret' method of creating annotations, there will likely be a significant time cost for a new user to annotate their data in order to achieve results similar to those shown. Importantly, this volume of required training per classification task suggests that (1) a user would need to know quite a bit about their system a priori and (2) the neural network 'only knows what it knows' and may have difficulty extrapolating to untrained cell types. Therefore, the authors' statement, 'MAPS can ... facilitate the discovery of novel biological processes' [Lines 57-59] may not be accurate.

The FNN presented appears to be of a simple, standard architecture. 'FNN' encompasses a broad category of neural networks that have been already used (and iterated and improved upon) in a wide variety of applications ranging from protein structure prediction [Refs. 1,2] to facial recognition software [Refs. 3,4]. More relevant to this manuscript, more optimized FNNs have been used in scRNA-seq-based [Refs. 5,6] and spatial-RNA-seq-based [Ref. 7] cell type classification, which fundamentally work the same way as MAPS, just with inputs of RNA expression matrices instead of protein. If the claim is, for example, that this FNN is generally better than existing neural networks specifically for the task of protein expression-based cell type classification, perhaps the authors can quantify the generalizability of their FNN to different datasets?

Finally, the 'Spatial' part of 'MAPS' seems to be misleading. The presented neural network does not incorporate spatial features at all (i.e. it knows nothing about the spatial relationships among cells, and does not, for example, use a neighboring cell's identity to assist in classifying the original cell). As mentioned above, MAPS currently appears to be functionally equivalent to existing scRNA-seq cell type classification paradigms in terms of input and output, which raises overall concerns about the significance of this work.

*** Other concerns:**

There is insufficient 'external validation' for the trained model in Fig. 1. The results in panels B-E are only from 'internal validation', i.e. data drawn from the same population as the training data. While internal validation is important to show, this data is inherently biased (e.g. by batch effects)

and therefore cannot independently evaluate the performance of the model. Notably, CRC-CODEX is briefly presented as a form of external validation in Fig. 2B. However, its F1 Score appears to be significantly lower than the internal validation F1 Scores (cHL 1 MIBI, cHL 2 MIBI, and cHL CODEX), which highlights the importance of presenting external validation for MAPS.

MAPS defaults to 500 epochs during training. It is fairly likely that this is causing the model to overfit the data, as 500 epochs is about an order of magnitude higher than typical. This point can likely be verified or disproved by analyzing the aforementioned external validation.

* Minor concern:

The authors mentioned that their code for reproducing their CellSighter and Astir results are available [Lines 647-649]. However, this reviewer was unable to find the aforementioned code.

* Suggestions:

It is highly suggested that external validation be presented as Fig. 1B-E instead of the current internal validation. This gives users a more realistic idea of the classification accuracy that MAPS can offer without significant retraining for new datasets. Please indicate which datasets were used for: training, validation ('internal validation'), and testing ('external validation') in the Figure legend or corresponding Methods section. One basic example of external validation is training on MIBI data with one or two patients' data removed, then using said data for testing.

Another suggestion would be to show the applicability and limitations of using a MAPS-trained model to classify new data. (e.g. At what point will a user have to retrain the model? Will the model hold if a user has the same or subset of markers but used a different fixative protocol, segmentation algorithm, or cohort with varying stages of lymphoma?)

If the significance of MAPS is the generalizability of a simple FNN to a variety of protein expression-based cell type classification tasks, then a suggestion would be to show high prediction accuracies when classifying other cancerous tissues of interest. Alternatively, it is possible to focus more on a model comparison or discuss methods to improve the neural network to reduce training burden while maintaining predictive power.

Due to the similarities between MAPS and existing scRNA-seq-based cell type classification mentioned above, it is suggested to put MAPS in context of these tools in the Introduction and/or Discussion sections.

* References

1. <https://pubmed.ncbi.nlm.nih.gov/32612753/>
2. <https://pubmed.ncbi.nlm.nih.gov/34265844/>
3. <https://pubmed.ncbi.nlm.nih.gov/29993957/>
4. <https://pubmed.ncbi.nlm.nih.gov/33201812/>
5. <https://pubmed.ncbi.nlm.nih.gov/33767197/>
6. <https://pubmed.ncbi.nlm.nih.gov/33817554/>
7. <https://pubmed.ncbi.nlm.nih.gov/34062119/>

RESPONSE TO REVIEWERS' COMMENTS

MAPS: Pathologist-level cell type annotation from tissue images through machine learning

We are grateful to the Editor and Reviewers for assessing this work and for their advice and positive feedback. In our revised manuscript, we have:

1. Expanded the manuscript from 2 Figures to 4 Figures, and also expanded upon the text to detail and clarify upon the capacity of MAPS to better answer the question raised by Reviewers. Provided more details in the manuscript or other resources over the training and using of the MAPS model, as suggested by Reviewer #1.
2. To answer the Reviewer #3's question, we applied the MAPS with subsets of the training data, and demonstrated its applicability over a smaller dataset. Also, we included an additional external dataset (DLBCL MIBI) to demonstrate the usability of the MAPS.
3. To answer Reviewers' concerns, we presented cross-dataset prediction results with MAPS and its counterparts, to demonstrate MAPS-trained model's applicability to new datasets.
4. Improved upon the general rewording, edited the text for clarification, updated the Figures and references as suggested by Reviewers #1, #2, and #3.

Reviewer #1:

Remarks to the Author:

MAPS is a neural network-based algorithm trained on “ground-truth” annotated samples (samples annotated by clustering and marker-based, pathologist-confirmed adjudication) that exhibits an impressive improvement on existing algorithms for cell type annotation of spatial data. Some questions raised by my reading that could strengthen or clarify the manuscript:

Training MAPS:

-Could cyTOF data be used crossfunctionally to train MAPS for MIBI classification within a given tissue type?

We thank the Reviewer for the time and effort in helping us strengthen this study. The kind of experiments the Reviewer requested will require the exact matched single cell dataset and imaging dataset of the same tissues, which is not available to date. Unfortunately, addressing this question is outside the scope of this work.

But nevertheless, we would like to point the Reviewer to other works published with some of the co-authors (Chen et al. 2023; Zhu et al. 2023). Those works achieve the cross functionally matching of single cell data to imaging data, although they are not related to the prediction of the cell type, which is the focus of this manuscript.

We additionally tested the performance of MAPS on a CyTOF dataset (training/validation: CyTOF; testing: CyTOF), the result shows a very high accuracy, which demonstrated the prediction ability of the MAPS on other modalities (Figure R1).

Figure R1, Confusion matrix of cell classification in CyTOF dataset.

Chen, S., Zhu, B., Huang, S. et al. Integration of spatial and single-cell data across modalities with weakly linked features. *Nat Biotechnol* (2023). <https://doi.org/10.1038/s41587-023-01935-0>

Zhu, B., Chen, S., Bai, Y., Chen, H., Liao, G., Mukherjee, N., ... & Jiang, S. (2023). Robust single-cell matching and multimodal analysis using shared and distinct features. *Nature Methods*, 20(2), 304-315.

-Could cell annotation from databases or user-inputted cell protein markers be used to train MAPS?

We presented two MIBI datasets on classic Hodgkin Lymphoma “cHL1” and “cHL2”, from different sample sources. To in part address the Reviewer’s suggestion, we applied the model trained with the dataset “cHL1”, towards “cHL2. Our results (Figure 3B) show lower but reasonable performance and also out-performed other cell phenotyping models.

Figure R2, Cross dataset performance comparison of MAPS model with ASTIR and CellSighter. Each bar indicates the average results of models (one for each fold) trained on cHL1 (MIBI) and evaluated on the whole cHL2 (MIBI) dataset. Error bars indicate the ± 1 standard deviation.

Figure 3B, Cross dataset performance comparison of MAPS model with ASTIR and CellSighter at class level. Each bar indicates the average results of models (one for each fold) trained on cHL1 (MIBI) and evaluated on the whole cHL2 (MIBI) dataset. Error bars indicate the ± 1 standard deviation.

As the Reviewer is well aware, the batch to batch variation, and imaging modality difference could all contribute to variation and deviation from an accurate annotation result. Additionally, such a robust spatial proteomics cell annotation database is not available yet, but our results

suggest that it may be used in part to better train MAPS.

Using MAPS:

-How will the algorithm be trained by users for new tissues (ie, not lymphoma or LN samples)? Will a training set be provided, or do training, ground-truth data need to be created de novo? On this note, when a reliable reference/training dataset is not available, could MAPS as an alternative simply output a graphical cell clustering with protein markers for each cluster instead of annotation? Related to this, could the algorithm provide as output the markers that define each cell type?

We thank the Reviewer for asking those questions. We have outlined this in Supplementary Figure 1B, and provided a detailed tutorial as part of the GitHub repository (https://github.com/mahmoodlab/MAPS/blob/main/tutorial/cell_phenotyping.ipynb), and also further detailed it in the METHODS. But in short:

1. For new tissue types, a set of pre-defined ground-truth for the new tissue type will be generated (consisting of 10-20% of the full dataset) to train the model before using the MAPS. These training/ground-truth data will be annotated de novo through iterative clustering and visual inspection with a board certified pathologist to achieve high accuracy, as detailed in our METHODS.
2. Should a training dataset not be available, (including the annotation of a subset of the data not performed), gradient attribution scores for cell phenotyping can be calculated using a pretrained model on new dataset, and the markers with high attribution scores can be recognized as the markers that define a cell phenotype. We caution users that these phenotype maps will need to be reviewed critically for accuracy based on *a priori* knowledge.

B Schematic of Workflow for Spatial Proteomics & MAPS

Figure S1B. Schematic of the workflow for spatial proteomics cell phenotyping accelerated by MAPS.

-Some cells are “unassigned”; could an “unassigned” threshold be set by the user, variably depending on the degree of confidence desired?

We thank the Reviewer for this question. We use the 'other' class for cells that do not fall under any predefined cell phenotype, or for rare cells that we are unable to confidently assign to a predefined cell phenotype. Given how MAPS outputs a probability per cell for each class, and assigns class labels based on highest probability, this allows users to further modify for increase/decrease the predefined threshold.

Misc:

-Were the reference (training) and test images the same? Presumably not, but important to clarify the details here.

We thank the Reviewer for pointing this out, we split each dataset into five-folds where four folds were used as training/validation (80%/20%) sets and the remaining fold was used as the test set, iteratively. We also make sure that the images of each case fall in only one fold. We have modified parts of the manuscript to better clarify this point.

Muhammad, Bai, Qiu, et al.

-Figure 2A legend -> why are there 2 colors for “Both correct”?

We thank the Reviewer for pointing this out, and the typo has been corrected.

Reviewer #2:

Remarks to the Author:

In the manuscript entitled “MAPS: Pathologist-level cell type annotation from tissue images through machine learning” by Muhammad Shaban et al. authors describe Machine learning for Analysis of Proteomics in Spatial biology (MAPS), a new machine learning approach for pathologist-level accuracy in cell annotation from spatial proteomics data, facilitating the rapid and precise cell type identification.

This work demonstrates that the proposed method outperforms existing state-of-the-art methods in terms of both accuracy and computational efficiency while showing cross-platform compatibility (MIBI and the novel CODEX).

The proposed method holds significant potential to expedite advances in tissue biology and the understanding complex biological systems i.e. in the context of solid tumors.

We thank the Reviewer for the kind words of support.

This work is innovative, the data interpretation and the solidity of novel method are well supported. I believe the work could meet the standards of Nature Communications after addressing the following revisions:

1. I suggest the author consider removing the acronym from the title

We thank the Reviewer for pointing this out, after discussion among the authors, we respectfully decided not to remove the acronym, in part to enhance discoverability and clarity in the ever-growing landscape of academic literature and methods. It's worth noting that many famous papers in the field of machine learning have employed acronyms in their titles for examples:

- **BERT:** Pre-training of Deep Bidirectional Transformers for Language Understanding
- **U-Net:** Convolutional Networks for Biomedical Image Segmentation
- **CellSighter:** a neural network to classify cells in highly multiplexed images

Devlin, J., Chang, M. W., Lee, K., & Toutanova, K. (2018). Bert: Pre-training of deep bidirectional transformers for language understanding. arXiv preprint arXiv:1810.04805.

Ronneberger, O., Fischer, P., & Brox, T. (2015). U-net: Convolutional networks for biomedical image segmentation. In Medical Image Computing and Computer-Assisted Intervention–MICCAI 2015: 18th International Conference, Munich, Germany, October 5-9, 2015, Proceedings, Part III 18 (pp. 234-241). Springer International Publishing.

Amitay, Y., Bussi, Y., Feinstein, B., Bagon, S., Milo, I., & Keren, L. (2023). CellSighter: a neural network to classify cells in highly multiplexed images. Nature communications, 14(1), 4302.

2. The font size for the cell type legend in Supplementary Figure 1A and S2B could be increased

We thank the Reviewer for pointing this out, we have modified the Figures and modified corresponding legends according to this guideline.

3. The introduction section is poor; no explanation is given to MIBI and CODEX, and no state-of-the-art context is provided. Nature Communications is not a sector journal, and due to the high level of the journal, much more attention to details should be provided in guiding also non-specialized readers. The authors are requested to make an extra effort in shaping the context of this innovative work in relation to the two platforms used and the state of the art of other approaches.

We thank the Reviewer for pointing this out, we originally intended this manuscript as a brief communication, but have since significantly expanded the INTRODUCTION part of the manuscript to provide better context of the work and state-of-art,

From:

“Recent advances in high-plex spatial proteomics have facilitated the simultaneous imaging of over 50 markers, thereby offering invaluable insights into protein expression and distribution within cellular and tissue architectures for phenotypic and functional investigations (Hickey et al. 2022; Pourmaleki et al. 2023).”

To:

“Compared to conventional low-plex imaging methods, recent advances in high-plex spatial proteomics techniques, such as MIBI, CODEX, cycIF, and IMC, allows for interrogation of 40-60 proteomic markers within a single tissue section (Angelo et al. 2014; Giesen et al. 2014; Lin et al. 2015; Goltsev et al. 2018; Keren et al. 2018; Keren et al. 2019; Schürch et al. 2020). These approaches offer invaluable insights into protein expression and distribution within cellular and tissue architectures for phenotypic and functional investigations, and are broadly applicable to fields such as cancer-immune and host-pathogen interactions (Hickey et al. 2022; Jiang et al. 2022; Pourmaleki et al. 2023). Iterative cyclical methods, such as CODEX and cycIF, achieve multiplexity through multiple cycles of staining, imaging, and bleaching/stripping of the labeling molecules using fluorescent microscopy methods and off-the-shelf reagents. However, these approaches may face barriers related to tissue degradation, difficulties in image registration, and epitopes loss during the cycling process. Mass spectroscopy-based methods, such as MIBI and IMC, are a different imaging modality requiring specialized instruments and custom conjugation of antibodies with isotopes. In

this case, all the markers may be acquired simultaneously to directly reconstruct multiplexed images for downstream analysis.

While these highly multiplexed images may provide new insights into biological processes, they also pose challenges in data processing, including the need for automated pipelines to distill the information from each single cell.”

From:

“However, these approaches may be lower in accuracy, or be computationally expensive, requiring more memory and taking longer times to train and infer.”

To:

“Geuenich et al. introduced ASTIR, an automated method for assigning cell identities using single-cell multiplexed imaging data. ASTIR utilizes a probabilistic model that incorporates prior knowledge of marker proteins to categorize cells into specific cell types. Amitay et al. introduced CellSighter, a cell classification pipeline based on deep learning. This method exhibits promising classification performance. However, it is noted for its shortcomings in computational efficiency due to its reliance on an ensemble of ten ResNet50-based models with random initializations, which can be computationally intensive and potentially resource-consuming.”

4. By rereading the discussion and conclusion, there is a lack of ability to go in-depth on the innovative points of the work contextualizing in relation to the selected models, to highlight the value of the work better, I suggest increasing a more detailed and point-specific discussion.

We thank the Reviewer for pointing this out, we have now rewritten the entire DISCUSSION section of the manuscript to provide better context of the work.

Reviewer #3:

Remarks to the Author:

* Summary:

The authors present MAPS, a Python package for cell type classification using a feed-forward neural network. They demonstrate the ability to discern over a dozen cell types from in-house datasets of canonical Hodgkin's Lymphoma using a feature set of 49 expression markers and a 50th segmented cell size feature, with high class precision rates ranging from 0.83 to 0.98. MAPS has the advantages of being faster and more accurate than previously published cell type classification tools such as CellSighter and Astir, and the code is open-source with an easy-to-follow vignette on the authors' Github page.

We thank the Reviewer for the time and suggestions.

* Primary concern:

It is of this reviewer's opinion that the significance of this work is unclear, for the following reasons.

While the FNN itself runs faster than its competitors, the authors' training data consists of more than 1 million cells. Even with the authors' 'cluster and interpret' method of creating annotations, there will likely be a significant time cost for a new user to annotate their data in order to achieve results similar to those shown. Importantly, this volume of required training per classification task suggests that (1) a user would need to know quite a bit about their system a priori and (2) the neural network 'only knows what it knows' and may have difficulty extrapolating to untrained cell types. Therefore, the authors' statement, 'MAPS can ... facilitate the discovery of novel biological processes' [Lines 57-59] may not be accurate.

We sincerely appreciate the Reviewer for raising the important concern regarding the time cost of annotating a large dataset for training in MAPS, which is exactly the problem we were hoping to solve. However, to attempt to solve this problem, we indeed painstakingly generated this ground truth through iterative "cluster and interpret".

To demonstrate how MAPS scales with dataset size and performance, we have provided results for different MAPS models trained using varying percentages of the training data, from 100% down to 5% (Figure 4A, Figure R3). Our data shows robust performance across the cell types for as low as 5% of the original training data size, and provides the users the opportunity to evaluate their choice of the appropriate training dataset size that aligns with their specific needs and available resources. It is also important to note that the curation of the training set is a one-time task, and the inference time cost is recurring. Therefore, having a faster model with superior performance, as demonstrated by MAPS, can result in substantial long-term benefits by reducing the time and resources required for future analyses.

Regarding the statement about MAPS facilitating the discovery of novel biological processes, we understand the concern, and we have now revised the language to better reflect the potential and limitations of the method in the text:

From:

“MAPS can significantly enhance our understanding of complex biological systems and facilitate the discovery of novel biological processes in situ.”

To:

“MAPS can significantly facilitate both the speed and quality of cell annotation process so that researchers can allocate more downstream efforts in unveiling novel biological processes in situ.”

Figure 4A, Precision, recall, and F1 scores of MAPS when trained with 5%, 10%, 25%, 50%, 75%, and 100% of the cHL1 training dataset and tested on the cHL1 testing set. Error bars represent ± 1 standard deviation.

Figure R3, Performance comparison of different MAPS models trained using different percentages of training sets from cHL1_MIBI dataset. Each bar indicates the average F1-score on test sets across five folds whereas error bars indicate the ± 1 standard deviation.

The FNN presented appears to be of a simple, standard architecture. 'FNN' encompasses a broad category of neural networks that have been already used (and iterated and improved upon) in a wide variety of applications ranging from protein structure prediction [Refs. 1,2] to facial recognition software [Refs. 3,4]. More relevant to this manuscript, more optimized FNNs have been used in scRNA-seq-based [Refs. 5,6] and spatial-RNA-seq-based [Ref. 7] cell type classification, which fundamentally work the same way as MAPS, just with inputs of RNA expression matrices of instead of protein. If the claim is, for example, that this FNN is generally better than existing neural networks specifically for the task of protein expression-based cell type classification, perhaps the authors can quantify the generalizability of their FNN to different datasets?

We appreciate the Reviewer's recognition of the similarity in network architecture with existing methods used for scRNA-seq cell type classification paradigms. Indeed, similar neural network architectures are frequently employed in various applied fields. However, it is crucial to clarify that MAPS was purposefully designed as a specialized tool for cell phenotyping in multiplexed imaging datasets, representing a distinct and unique domain of application. Our primary goal was to demonstrate the practicality and efficacy of MAPS within this specific domain and to showcase its superior performance when compared to established methods like Astir and

CellSighter, within the context of multiplexed imaging. We have now provided the performance comparison MAPS with ASTIR and CellSighter on 5 different datasets (cHL1 MIBI, cHL2 MIBI, cHL CODEX, DLBCL MIBI, CRC CODEX). We also compared the performance of the MAPS model with ASTIR and CellSighter models trained on the cHL1 MIBI dataset and evaluated on the cHL2 MIBI dataset (Figure R4, Figure 3B).

Figure R4, Cross dataset performance comparison of MAPS model with ASTIR and CellSighter. Each bar indicates the average results of models (one for each fold) trained on cHL1 (MIBI) and evaluated on the whole cHL2 (MIBI) dataset. Error bars indicate the ± 1 standard deviation.

Figure 3B, Cross dataset performance comparison of MAPS model with ASTIR and CellSighter at class level. Each bar indicates the average results of models (one for each fold) trained on cHL1 (MIBI) and evaluated on the whole cHL2 (MIBI) dataset. Error bars indicate the ± 1 standard deviation.

Finally, the 'Spatial' part of 'MAPS' seems to be misleading. The presented neural network does not incorporate spatial features at all (i.e. it knows nothing about the spatial relationships among cells, and does not, for example, use a neighboring cell's identity to assist in classifying the original cell). As mentioned above, MAPS currently appears to be functionally equivalent to existing scRNA-seq cell type classification paradigms in terms of input and output, which raises overall concerns about the significance of this work.

We acknowledge that the current version of the MAPS algorithm does not incorporate spatial features, a step that may be incorporated into future iterations. Importantly, we highlight that despite this limitation, MAPS outperformed its counterpart, CellSighter, which relies on spatial neighborhood information but incurs high computation and memory costs.

* Other concerns:

There is insufficient 'external validation' for the trained model in Fig. 1. The results in panels B-E are only from 'internal validation', i.e. data drawn from the same population as the training data. While internal validation is important to show, this data is inherently biased (e.g. by batch effects) and therefore cannot independently evaluate the performance of the model. Notably, CRC-CODEX is briefly presented as a form of external validation in Fig. 2B. However, its F1 Score appears to be significantly lower than the internal validation F1 Scores (cHL 1 MIBI, cHL 2 MIBI, and cHL CODEX), which highlights the importance of presenting external validation for MAPS.

We thank the reviewer for highlighting the importance of external validation and suggesting a simple way for external validation: "One basic example of external validation is training on MIBI data with one or two patients' data removed, then using said data for testing". We have now adapted the suggested external evaluation method in our experimental setup and regenerated all the results of ASTIR, CellSighter, and MAPS on cHL1 MIBI, cHL2 MIBI, cHL CODEX, and a new DLBCL MIBI dataset. Specifically, we split all datasets into 5 folds while making sure that cells from each case only fall in one of the folds. Then we iteratively used four folds for model training and validation (optimal model selection), and the remaining fold for testing. The results indicate that MAPS remains the best performing method with a high F1-score (Figure 2B, Left).

Figure 2B, (Left) Performance comparison of MAPS model with two other cell phenotyping models using F1-score on 4 different datasets. Each bar indicates the average F1-score on test sets across five folds whereas error bars indicate the ± 1 standard deviation.

MAPS defaults to 500 epochs during training. It is fairly likely that this is causing the model to overfit the data, as 500 epochs is about an order of magnitude higher than typical. This point can likely be verified or disproved by analyzing the aforementioned external validation.

We acknowledge the Reviewer's concern regarding overfitting as the default setting of 500 epochs is perceived as high. However, we would like to clarify that the number of epochs (num_epochs) is a hyperparameter that we carefully adjust for different datasets based on their size and complexity as explained in the **Training details** section of METHODS in the revised manuscript as follows:

“For larger datasets (cHL 1 MIBI), the number of epochs is set lower, as more training steps are performed within each epoch. Conversely, for smaller datasets (cHL 2 MIBI and cHL CODEX), we utilize a higher number of epochs to ensure an adequate number of training

steps. Specifically, the model was trained for 100 epochs on the cHL 1 (MIBI) dataset, and for 500 epochs on all other datasets. Additionally, we implement two essential hyperparameters, namely minimum epoch and patience, specifically designed to address the issue of overfitting. The minimum epoch ensures that the model undergoes a minimum number of training epochs, and the patience parameter enables early stopping if the validation performance does not improve, thus mitigating the risk of overfitting. The model with lowest validation loss was selected as the best model for evaluation on test sets and inference on new data.”

* Minor concern:

The authors mentioned that their code for reproducing their CellSighter and Astir results are available [Lines 647-649]. However, this reviewer was unable to find the aforementioned code.

We thank the Reviewer for pointing this out, we now have included the codes we used to reproduce CellSighter and Astir results in the public GitHub repository (<https://github.com/mahmoodlab/MAPS>).

* Suggestions:

It is highly suggested that external validation be presented as Fig. 1B-E instead of the current internal validation. This gives users a more realistic idea of the classification accuracy that MAPS can offer without significant retraining for new datasets. Please indicate which datasets were used for: training, validation ('internal validation'), and testing ('external validation') in the Figure legend or corresponding Methods section. One basic example of external validation is training on MIBI data with one or two patients' data removed, then using said data for testing.

We thank the Reviewer for the valuable suggestion about how to report results of external validation. As mentioned above, we have now adapted the suggested external evaluation method in our experimental setup and regenerated all the results of ASTIR, CellSighter, and MAPS on cHL1 MIBI, cHL2 MIBI, cHL CODEX, and DLBCL MIBI datasets.

Figure 2B, Comparisons of precision, recall, and F1 score among ASTIR, CellSighter, and MAPS, across three in-house datasets (cHL1 MIBI, cHL2 MIBI, cHL CODEX) and an external dataset DLBCL MIBI. Error bars represent ± 1 standard deviation.

Another suggestion would be to show the applicability and limitations of using a MAPS-trained model to classify new data. (e.g. At what point will a user have to retrain the model? Will the model hold if a user has the same or subset of markers but used a different fixative protocol, segmentation algorithm, or cohort with varying stages of lymphoma?)

We thank the Reviewer for suggesting a way to gauge the applicability of MAPS to new data. We applied the MAPS, ASTIR, and CellSighter models trained on cHL1 to predict cell phenotypes of cHL2. Overall, all three models showed a decrease in performance, with the performance of MAPS and CellSighter still acceptable for most of the cell types. However, due to the heterogeneity among different datasets and the lack of established baseline for proteomics data, which, in turn, causes difficulties to adjust for the heterogeneity, a decrease in performance was expected.

Figure R5, Cross dataset performance comparison of MAPS model with ASTIR and CellSighter. Each bar indicates the average results of models (one for each fold) trained on cHL1 (MIBI) and evaluated on the whole cHL2 (MIBI) dataset. Error bars indicate the ± 1 standard deviation.

Figure 3B, Cross dataset performance comparison of MAPS model with ASTIR and CellSighter at class level. Each bar indicates the average results of models (one for each fold) trained on cHL1 (MIBI) and evaluated on the whole cHL2 (MIBI) dataset. Error bars indicate the ± 1 standard deviation.

If the significance of MAPS is the generalizability of a simple FNN to a variety of protein

expression-based cell type classification tasks, then a suggestion would be to show high prediction accuracies when classifying other cancerous tissues of interest. Alternatively, it is possible to focus more on a model comparison or discuss methods to improve the neural network to reduce training burden while maintaining predictive power.

We greatly appreciate the Reviewer's insightful comments and suggestions. MAPS' significance is two-fold.

Firstly, MAPS' applicability across a diverse range of spatial proteomic datasets in terms of tissue type and disease model, coupled with its consistently high performance. We have now reported results on another external dataset (DLBCL) where MAPS again outperformed its counterparts.

Secondly, the lightweight nature of MAPS, owing to its straightforward architecture, coupled with its rapid training and inference times, is a critical feature. This efficiency sets MAPS apart from other methods, making it a valuable tool for large-scale spatial proteomics analyses where computational resources are a critical consideration.

Figure S2B, (Bottom left) Performance comparison of MAPS model with ASTIR and CellSighter at class level on an external DLBCL dataset. Each bar indicates the average results of models (one for each fold) whereas error bars indicate the ± 1 standard deviation.

Due to the similarities between MAPS and existing scRNA-seq-based cell type classification mentioned above, it is suggested to put MAPS in context of these tools in the Introduction and/or Discussion sections.

We appreciate the Reviewer's comment. As detailed above, we want to emphasize that although similar architectures are frequently studied and applied in the single cell analysis realm, such as scRNA-seq. Given that MAPS is designed for cell phenotyping in multiplexed imaging datasets, this represents a distinct and unique domain of application. Therefore, we put MAPS in the context of established methods for multiplex image analysis, such as Astir and CellSighter, in the introduction section to keep the manuscript more cohesive.

Related text in INTRODUCTION:

“Promising automated approaches developed recently include probabilistic inferential approaches (Geuenich et al. 2021; Zhang et al. 2022), and convolutional neural networks (Amitay et al. 2022; Brbić et al. 2022). Geuenich et al. introduced ASTIR, an automated method for assigning cell identities using single-cell multiplexed imaging data. ASTIR utilizes a probabilistic model that incorporates prior knowledge of marker proteins to categorize cells into specific cell types. Amitay et al. introduced CellSighter, a cell classification pipeline based on deep learning. This method exhibits promising classification performance. However, it is noted for its shortcomings in computational efficiency due to its reliance on an ensemble of ten ResNet50-based models with random initializations, which can be computationally intensive and potentially resource-consuming.”

* References

1. <https://pubmed.ncbi.nlm.nih.gov/32612753/>
2. <https://pubmed.ncbi.nlm.nih.gov/34265844/>
3. <https://pubmed.ncbi.nlm.nih.gov/29993957/>
4. <https://pubmed.ncbi.nlm.nih.gov/33201812/>
5. <https://pubmed.ncbi.nlm.nih.gov/33767197/>
6. <https://pubmed.ncbi.nlm.nih.gov/33817554/>
7. <https://pubmed.ncbi.nlm.nih.gov/34062119/>

REVIEWERS' COMMENTS

Reviewer #1 (Remarks to the Author):

The authors in their revision have further clarified the practical approach to use of MAPS and have also addressed my suggestion for exploring cross-dataset training with informative and potentially useful results. I remain of the opinion that MAPS with its impressive relative performance in cell type annotation of spatial data will be a useful tool.

Reviewer #2 (Remarks to the Author):

The authors well answered to all concerns previously raised. I am satisfied with the new version of the manuscript, I suggest it's publication in Nature Communications.

Reviewer #3 (Remarks to the Author):

This reviewer thanks the authors for taking the given feedback into consideration in their revision. It is clear that since the initial submission, the authors have put in significant time and effort to present more definitive literary context and rigorous quantitative data, particularly in regard to external validation.

The authors acknowledge in their rebuttal that MAPS does not yet incorporate spatial features, which, in this reviewer's opinion, remains a bit of an oversight. Nonetheless, this reviewer looks forward to seeing how the authors implement the 'S' in MAPS in future iterations of their work.